# Effects of Strain Differences, Humidity Changes, and Saliva Contamination on the Inactivation of SARS-CoV-2 by Ion Irradiation

**DOI:** 10.3390/v16040520

**Published:** 2024-03-28

**Authors:** Afifah Fatimah Azzahra Ahmad Wadi, Daichi Onomura, Hirokazu Funamori, Mst Mahmuda Khatun, Shunpei Okada, Hisashi Iizasa, Hironori Yoshiyama

**Affiliations:** 1Department of Microbiology, Faculty of Medicine, Shimane University, 89-1 Enya, Izumo 693-8504, Shimane, Japan; m229401@med.shimane-u.ac.jp (A.F.A.A.W.); mahmuda.biochemistry@gmail.com (M.M.K.); s.okada@med.shimane-u.ac.jp (S.O.); iizasah@med.shimane-u.ac.jp (H.I.); 2Faculty of Medicine, University of Muslim Indonesia, Makassar 9023, South Sulawesi, Indonesia; 3Division of Virology, Department of Infection and Immunity, Faculty of Medicine, Jichi Medical University, Shimotsuke 329-0498, Tochigi, Japan; onomura.daichi@jichi.ac.jp; 4Sharp Corporation, Yao 581-8585, Osaka, Japan; funamori.hirokazu@sharp.co.jp

**Keywords:** ion, SARS-CoV-2, inactivation, VOC, saliva, humidity

## Abstract

One of the methods to inactivate viruses is to denature viral proteins using released ions. However, there have been no reports detailing the effects of changes in humidity or contamination with body fluids on the inactivation of viruses. This study investigated the effects of humidity changes and saliva contamination on the efficacy of SARS-CoV-2 inactivation with ions using multiple viral strains. Virus solutions with different infectious titers were dropped onto a circular nitrocellulose membrane and irradiated with ions from 10 cm above the membrane. After the irradiation of ions for 60, 90, and 120 min, changes in viral infectious titers were measured. The effect of ions on virus inactivation under different humidity conditions was also examined using virus solutions containing 90% mixtures of saliva collected from 10 people. A decrease in viral infectivity was observed over time for all strains, but ion irradiation further accelerated the decrease in viral infectivity. Ion irradiation can inactivate all viral strains, but at 80% humidity, the effect did not appear until 90 min after irradiation. The presence of saliva protected the virus from drying and maintained infectiousness for a longer period compared with no saliva. In particular, the Omicron strain retained its infectivity titer longer than the other strains. Ion irradiation demonstrated a consistent reduction in the number of infectious viruses when compared to the control across varying levels of humidity and irradiation periods. This underscores the notable effectiveness of irradiation, even when the reduction effect is as modest as 50%, thereby emphasizing its crucial role in mitigating the rapid dissemination of SARS-CoV-2.

## 1. Introduction

Severe acute respiratory syndrome coronavirus 2 (SARS-CoV-2) is characterized by a positive-sense, single-stranded RNA genome comprising approximately 30,000 nucleotides, and is responsible for coronavirus disease 2019 (COVID-19) [1]. The SARS-CoV-2 virion contains structural proteins such as spike (S), envelope (E), membrane (M), and nucleocapsid (N). The S protein is used for viral cell entry through interactions with angiotensin-converting enzyme 2 (ACE2) [2]. Coronavirus NSP14 has 3′-5′ exonuclease activity, and its RNA proofreading mechanism prevents the viral genome from mutation [3]. However, SARS-CoV-2 frequently generates mutant viruses because of its extremely high replication frequency and transmissibility. These characteristics have induced the emergence of new variants distinct in terms of transmission, pathogenicity, diagnostics, and vaccine efficacy from the original virus that began circulating in China in December 2019. Mutant viruses that cause clinical and public health concerns are called variants of concern (VOCs). Public health officials need to identify and monitor VOCs to effectively control their spread [4,5].

The development of antiviral agents and vaccines has been intensively pursued to control the spread of this virus [6,7]. However, because of the frequent mutations of the S gene in VOCs, new VOCs rapidly emerge after vaccination [5]. 

The physical and chemical inactivation of the virus is an effective alternative method for controlling the spread of SARS-CoV-2 [8]. A variety of disinfectants, such as sodium hypochlorite, hydrogen peroxide, alcohols, povidone-iodine, glutaraldehyde, ultraviolet radiation, and dry or moist heat, are used for disinfection. The direct inactivation of SARS-CoV-2 proteins and RNA is effective against all VOCs [9]. One method for inactivating microorganisms in the environment is the use of ions that are less harmful to the human body [10,11]. When a direct-current voltage is applied to an ion generator, a discharge phenomenon occurs at the end of the electrode, producing both ions and ozone [12]. Ions are derived from water and oxygen molecules in the air [13,14,15].

The efficiency of virus inactivation by ions is influenced by the indoor environment. Indoor relative humidity is expected to modulate the occurrence of COVID-19 [16]. Moderate humidity has been reported to have an association with COVID-19 outbreaks [17]. We considered ionization as a means to disinfect viruses in the environment. However, ions undergo chemical reactions upon contact with water. Because intermediate humidity is reported to have an association with COVID-19 outbreaks [18], it is necessary to investigate how humidity affects the effectiveness of ion irradiation against SARS-CoV-2.

Another factor affecting the transmission of airborne viruses is the presence of saliva. SARS-CoV-2 is transmitted through the eyes, nose, and mouth via microdroplets produced from saliva [1,18]. For example, artificial saliva has shown protective effects against coronaviruses suspended in indoor air, especially at low relative humidity [19]. However, saliva secreted from the human body contains not only sugar, but also proteases that degrade viral proteins, and it remains unclear how saliva affects the effectiveness of ion irradiation against SARS-CoV-2. Moreover, little information is available regarding the extent to which changes in humidity and body fluid contamination affect SARS-CoV-2 inactivation. In this study, we investigated the effects of ion irradiation on SARS-CoV-2 VOCs deposited on materials at various humidity levels in the presence of saliva.

## 2. Materials and Methods

### 2.1. Cell Cultures

VeroE6/transmembrane protease serine 2 (TMPRSS) cells were cultured at 37 °C in 5% CO_2_ in Dulbecco’s modified Eagle medium (DMEM)/Ham’s F-12 medium (FUJIFILM Wako Pure Chemical Corp, Osaka, Japan) supplemented with 5% fetal bovine serum (FBS) (Sigma-Aldrich, St. Louis, MO, USA) and 1% penicillin (PC)–streptomycin (SM) mixed solution (Nacalai tesque, Kyoto, Japan). Moreover, a VeroE6/TMPRSS2 culture was maintained with the medium containing 1 mg/mL of G418 (Promega, Madison, WI, USA) to maintain the expression of TMPRSS2 [20]. However, VeroE6/TMPRSS2 cells were passaged twice with the G418-free medium before infection with SARS-CoV-2. Then, 50 µL of cell suspension that contained 2 × 10^4^ VeroE6/TMPRSS2 cells was plated on each well of a 96-well culture plate and incubated for 24 h at 37 °C using a 5% CO_2_ incubator.

### 2.2. Viruses

The SARS-CoV-2 Wuhan strain (Pango lineage: A, hCoV-19/Japan/TY/WK-521/2019), as well as the Alpha (Pango lineage: B.1.1.7, hCoV-19/Japan/QK002/2020), Delta (Pango lineage: B.1.617.2, hCoV-19/Japan/TY11-927/2021), and Omicron (Pango lineage: BA.1, hCoV-19/Japan/TY38-873/2021) variants were provided by the national institute of infectious diseases (NIID, Tokyo, Japan). Each virus was expanded in VeroE6/TMPRSS2 cells and culture supernatants were stored at −80 °C. The SARS-CoV-2 experiments were approved by the Biological Safety Committee of Shimane University and conducted in a biosafety level three (BSL-3) laboratory at the Faculty of Medicine, Shimane University. The viral TCID_50_ titers used in the assay are expressed with the mean plus–minus deviation in parentheses as follows: the Wuhan strain, at 30% humidity (1 × 10^5.5^ ± 1 × 10^0.1^), 60% humidity (1 × 10^5.9^ ± 1 × 10^0.3^), and 80% humidity (1 × 10^5.7^ ± 1 × 10^0.5^); the Alpha strain, at 30% humidity (1 × 10^5.7^ ± 1 × 10^0.5^) and 60% humidity (1 × 10^6.0^ ± 1 × 10^0.4^); the Delta strain, at 30% humidity (1 × 10^4.8^ ± 1 × 10^0.4^), 60% humidity (1 × 10^6.6^ ± 1 × 10^0.5^), and 80% humidity (1 × 10^6.8^ ± 1 × 10^0.4^); and the Omicron strain, at 30% humidity (1 × 10^6.4^ ± 1 × 10^0.1^), 60% humidity (1 × 10^6.4^ ± 1 × 10^0.1^), and 80% humidity (1 × 10^5.8^ ± 1 × 10^0.2^).

### 2.3. Virus Titration

In a 96-well plate, 2000 VeroE6/TMPRSS2 cells were seeded and cultured at 37 °C using a 5% CO_2_ incubator for 24 h in DMEM/Ham’s F-12 medium. The following day, 100 µL of 3.16-fold serial dilution of each virus in DMEM containing 1% PC/SM was added to each well of a 96-well plate. After 96 h of incubation with the virus, the cytopathic effect (CPE) was observed under a microscope. Four days post-infection, the cells were fixed with ethanol (FUJIFILM Wako, Tokyo, Japan) containing 16.7% acetic acid (FUJIFILM Wako). Each well was stained with 0.5% amido black (FUJIFILM Wako) dissolved in a solution containing 45% ethanol, 45% distilled water, and 10% acetic acid. Each well was determined to be positive or negative for viral infection. The 50% tissue culture infectious dose (TCID_50_) was determined by the Behrens–Karber method.

### 2.4. Saliva

Saliva was obtained from individuals who provided informed consent, following the ethical rules of Shimane University (20210426-3). Saliva samples were collected from 10 healthy individuals without SARS-CoV-2 infection. Collected saliva was mixed and centrifuged twice for 5 min at 200× *g*. The supernatant was filter-sterilized by being passed through a 0.45 μm filter membrane. The saliva was stored at −80 °C.

### 2.5. Treatment of Viruses with Ions

The experiments were conducted in a sealed humidity-controlled container (50–31–30 cm) with a built-in ion-generating electrode (Figure 1). First, 50 µL of each titrated virus solution was dropped onto a circular nitrocellulose membrane with a pore size of 0.025 µm and a diameter of 13 mm (Merk-Millipore, Billerica, MA, USA). An ion-emitting device (IZ-C80S2; Sharp Co., Ltd., Osaka, Japan) was subsequently placed 10 cm above the membrane and ion irradiation was applied for 60, 90, and 120 min. An ozone catalyst was incorporated into the air flow path to remove the antiviral effect of the ozone. In addition, the inactivating effect of ions on the virus was studied under different humidity conditions (30%, 60%, and 80%) using viral solutions containing 90% mixed saliva. A schematic image depicting the inactivating effect of ions with or without saliva is shown in Figure 2.

### 2.6. Humidity Conditions

A humidifier and dehumidifier were used to maintain the desired humidity levels for long periods. Humidity was controlled using a Humiai BLE-SD12-010 (Saginomiya, Tokyo, Japan). Ion irradiation was performed in containers maintained at humidity levels of 30%, 60%, and 80% (Figure 1). 

To set the 30% humidity condition, air conditioning in the BSL-3 laboratory was stopped and the dehumidifier was started in the evening of the day before the experiment. Four plastic cases containing silica gel were placed in a container. An air circulation system was configured to maintain a humidity level of 30% by operating a pump.

To prepare the 60% humidity condition, the humidity was adjusted based on the humidity on the day before the experiment. If the humidity level fell below 50%, a humidifier was used to adjust the humidity to 60%; if the humidity level exceeded 70%, the air conditioner was turned off and a humidifier was used to adjust the humidity level to 60%.

To prepare the 80% humidity condition, the humidifier was turned on and two plastic containers containing sponges soaked in sterile water were placed inside the test container. The humidity was maintained at 80% using a pump and air circulation system. 

Finally, the humidity of the container was checked on the day of the experiment to ensure that the specified humidity of ±4% was achieved for each experimental condition. Ozone concentration was measured by an ozone gas analyzer (EG-700EIII; EBARA JITSUGYO, Tokyo, Japan).

### 2.7. Statistical Analysis

To compare the decrease in infectious titers among the different viral strains, relative values at 0 min were calculated and displayed as a log 10-fold change. Similar standardization of the infectivity titers was conducted across multiple strains to evaluate the effectiveness of ion irradiation in inactivation. The lower the bar, the higher the degree of viral inactivation (Figure 3 and Appendix A). 

Student’s *t*-test was used to analyze the statistical difference between the two independent groups. An ANOVA test was used to analyze the statistical differences among more than three independent groups. Moreover, statistically significant differences in the reduction in viral infectivity between each strain were evaluated using Tukey’s multiple comparison test. Data are expressed as the mean ± standard deviation (SD). Statistical significance was set at *p* < 0.05.

## 3. Results

### 3.1. Changes in Infectivity of Four VOCs as a Result of Ion Irradiation in 30% or 60% Humidity

In an environment with 30% humidity, the virus was naturally inactivated over time, and the viral infectious titers decreased. However, the natural inactivation of the Omicron strain, one of the SARS-CoV-2 VOCs, was lower than that of the other strains at 60 and 120 min (Figure 4a and Appendix A). 

Ion irradiation promoted viral inactivation over time (Figure 4a). The Wuhan and Alpha strains exhibited similar inactivation patterns after ion irradiation. For the Wuhan, Alpha, and Delta strains, inactivation by ion irradiation was the strongest at the earliest time point of 60 min, and the rate of inactivation decreased as the duration extended to 90 and 120 min. Conversely, the Omicron strain was inefficiently inactivated by ion irradiation at 60 min. However, the Omicron strain received similar inactivation by ion irradiation as the other three strains at 90 and 120 min (Figure 4a).

Similar to the case of 30% humidity, the viral infectivity titers decreased over time in the 60% humidity environment (Figure 4b). In contrast to 30% humidity, ion irradiation promoted a further decrease in infectivity for all strains after 90 min (Figure 4b). The inactivation of the Omicron strain by ion irradiation was weaker than that of Wuhan and Delta strains at 90 min; however, inactivation was stronger than that of the Wuhan and Alpha strains at 120 min (Figure 4b and Appendix A). In addition, the inactivation of the Omicron strain at 60% humidity showed no difference between with and without irradiation at 60 min and 90 min, but ion irradiation showed more than 100-fold reduction at 120 min (120 min without ion irradiation: 1 × 10^−1.3^; 120 min with ion irradiation: 1 × 10^−3.5^). On the other hand, ion irradiation at 30% humidity inactivated the Omicron strain after 60, 90, and 120 min of irradiation (60 min without ion irradiation: 1 × 10^−0.4^; 60 min with ion irradiation: 1 × 10^−1.0^; 90 min without ion irradiation: 1 × 10^−1.5^; 90 min with ion irradiation: 1 × 10^−3.8^; 120 min without ion irradiation: 1 × 10^−2.2^; 120 min with ion irradiation: 1 × 10^−4^) (Figure 4). Because the Omicron strain showed a distinctive inactivation pattern compared to the other three strains, we used the Wuhan strain as a representative of the Wuhan, Alpha, and Delta strains. Subsequent experiments were conducted using the Wuhan and Omicron strains.

### 3.2. Changes in Infectivity of the Wuhan and Omicron Strains as a Result of Ion Irradiation in 30%, 60%, and 80% Humidity Environments

The most pronounced inactivation of the Wuhan strain by ion irradiation at 80% humidity was observed after 240 min (Figure 5a). The inactivation of the Omicron strain by ion irradiation was lower at 80% humidity for 240 min than at 30% humidity for 90 min (Figure 5b). These results indicated that ion irradiation inactivated SARS-CoV-2; however, higher humidity levels increased the time required for inactivation. In addition, the Omicron strain was unique in its resistance to inactivation at high humidity. A heat map was created based on the average values of the results obtained in the experiments shown in Figure 4 and Figure 5 (Appendix A).

### 3.3. Changes in Infectivity of the Wuhan and Omicron Strains in Saliva at 30%, 60%, and 80% Humidity as a Result of Ion Irradiation

As saliva is important for the transmission of respiratory pathogens [21], the effect of ion irradiation on infectious SARS-CoV-2 in saliva was investigated. In the Wuhan strain, saliva containment reduced inactivation by ion irradiation to non-irradiation levels at 30% and 60% humidity. However, at 80% humidity, additional inactivation by ion irradiation was observed after 120, 180, and 240 min of ion irradiation (Figure 6a). In contrast, in the Omicron strain, the virus with saliva started inactivation by ion irradiation at an earlier time point (60 and 90 min), which was evident at 30% and 60% humidity (Figure 6b).

### 3.4. Changes in Infectivity of the Omicron Strain with or without Saliva as a Result of Ion Irradiation at 30%, 60%, and 80% Humidity

As the Omicron strain was more resistant to inactivation, we examined the effects of saliva content on the Omicron strain at 30%, 60%, and 80% humidity (Figure 5b and Figure 6b). As shown in Figure 5b, a more than 1000-fold reduction in infectivity of the Omicron strain was obtained for longer than 90 min at 30%, longer than 120 min at 60%, and longer than 180 min at 80% humidity. However, for contaminated saliva, more than 120 min of ion irradiation was required to reduce the viral infection titers to less than one thousandth of the initial inoculum at 30%, 60%, and 80% humidity (Figure 6b). These results indicate that ion irradiation reduced infectivity at 30%, 60%, and 80% humidity; however, saliva contamination attenuated the inactivation of the Omicron strain. Using Wuhan, Omicron, and Delta strains suspended in saliva, a heat map was created using the mean log 10-fold change in TCID_50_ per mL with and without ion exposure at 30%, 60%, and 80% humidity (Appendix A). This also includes the experimental data in Figure 6.

As discussed before, saliva content facilitated the inactivation of the Omicron strain without ion irradiation at 30%, 60%, and 80% humidity. Saliva itself has weak anti-viral activity; however, at the same time, saliva content reduced the inactivation by ion irradiation at 30%, 60%, and 80% humidity.

## 4. Discussion

Airborne and contact transmission are considered the main pathways for the spread of SARS-CoV-2. Similar to other respiratory viruses, high humidity is expected to enhance the transmission risk of SARS-CoV-2. However, increasing humidity was associated with declining COVID-19 cases in the spring and summer, while decreasing humidity and increasing residential mobility during the winter months caused an increase in COVID-19 cases [22]. It has been reported that the attenuation of SARS-CoV-2 due to temperature is minimal [23,24]. However, other studies have highlighted that coronavirus is stable at lower temperatures [25,26]. As indoor environmental conditions mostly affect the trend of the virus, it is necessary to investigate the attenuation of the virus under laboratory conditions. Moreover, we conducted experiments to clarify the effects of a mixture of humidity and saliva on viral attenuation.

In cases of infectious disease outbreaks, ultraviolet (UV) light is commonly utilized as an effective method for deactivating viruses in the environment. However, due to the potential risks of causing harm to the skin and eyes, we deemed it necessary to explore the efficacy of ion irradiation. By integrating ion irradiation devices into air conditioning systems, there is a possibility that viruses in the air could be deactivated as air circulates. Furthermore, ion irradiation can also deactivate viruses adhering to the surfaces of objects. Particularly in high-risk environments such as specific rooms or facilities with elevated infection risks, we believe that regular ion irradiation could significantly reduce the infectious viral load in the air, thus providing a high level of effectiveness in preventing SARS-CoV-2 infections.

Natural decay over time was observed for all the VOCs. Ion irradiation further accelerated the reduction in viral infectivity. The rate of decrease in viral infectivity was the fastest when ion irradiation was continued for 60 min at 30% humidity and continually decreased thereafter, reaching a maximum reduction at 90 min (Figure 4a). In contrast, the maximum decrease at 60% humidity required 120 min of ion irradiation (Figure 4b). Moreover, the rate of decrease in infectivity owing to ion irradiation was lower at 80% humidity than at 30% or 60% humidity (Figure 5).

The inactivation of viruses in saliva provides important information for controlling viral infections, as SARS-CoV-2 has been detected in collected saliva at a 91.7% frequency [27]. Artificial saliva has been shown to have a strong protective effect against coronaviruses, which is more pronounced at low relative humidity [19]. When SARS-CoV-2 was present in the saliva, the rate of decrease in infectivity was slower than that of the virus suspended in media. In particular, we showed that saliva contamination strongly diminished the decrease in infectivity by ion irradiation in the Omicron strain compared to that in the Wuhan strain (Figure 6).

Drying appeared to play an important role in reducing viral infectivity, and ion irradiation accelerated the decrease in infectivity. The inactivation of viral infectivity by ion irradiation was effective for all VOCs. However, in an environment with 80% humidity, 90 min of ion irradiation was required to reduce the viral infectivity (Figure 5). Saliva contamination prolonged the time required to reduce infectivity, presumably by protecting the virus from drying out (Figure 6). Notably, the Omicron strain exhibited prolonged infectivity compared to the other strains when the saliva was contaminated (Figure 4, Figure 5 and Figure 6).

We have demonstrated that ions can inactivate viruses while removing as much ozone as possible. Because ozone is heavier than air, we attempted to minimize the effect of ozone by circulating air in the box and absorbing ozone by incorporating an ozone catalyst into the air flow path (Figure 1). By using the ozone catalyst, the ozone concentration around the virus-loaded membrane was mostly kept to 0.05 ppm at all humidity conditions and never exceeded 0.1 ppm (Appendix A).

As people spend most of their daily lives inside buildings, it is important to understand viral dynamics and find a strategy to prevent SARS-CoV-2 transmission in indoor environments [28]. The inactivation effect of these ions is stronger for many viral strains at high humidity. The Omicron strain was found to have a lower ion inactivation effect than the other three strains. A similar observation was reported by Hirose et al., who measured residual virus titers on plastic and human skin surfaces [29]. Additionally, it has been reported by another research group that the Omicron variant is less susceptible to inactivation compared to the Wuhan strain when adhered to stainless steel, polypropylene, glass, and paper [30]. The differences in properties that lead to the resistance of the Omicron variant to inactivation compared to other strains can be speculated to arise from mutations in the structural protein genes. The Omicron variant carries significant mutations in the spike protein, envelope protein, membrane protein, and nucleocapsid protein, which makes the Omicron strain markedly different from the other strains [30]. Particularly, amino acid mutations in the spike protein differ by as many as 30 sites from the Wuhan strain [30]. Moreover, the receptor-binding domain (RBD) of the spike protein, which in the Wuhan strain binds to ACE2 only partially, can bind to ACE2 entirely in the Omicron variant [31]. 

We found that viruses in the saliva are resistant to inactivation probably by protecting viruses from drying, but ions can bind viral surface proteins more efficiently at 30% and 60% humidity than at 80% humidity (Figure 2). However, it is important to note that saliva possesses a mild antiviral effect, as observed in Figure 6, likely attributed to the enzymes it contains.

In conclusion, ion irradiation effectively reduced the number of infectious viruses compared to the control at all humidity levels. However, the Omicron variant required longer (120 min) irradiation at 60% humidity compared to the other VOCs. Irradiation is extremely effective, even if the reduction effect is only 50%, highlighting its relevance in preventing the rapid spread of SARS-CoV-2.

## Figures and Tables

**Figure 1 viruses-16-00520-f001:**
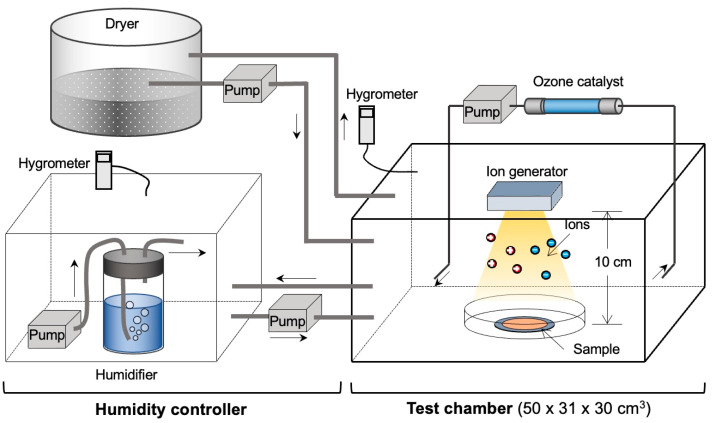
Virus inactivation test using ions in a sealed container with a humidity controller.

**Figure 2 viruses-16-00520-f002:**
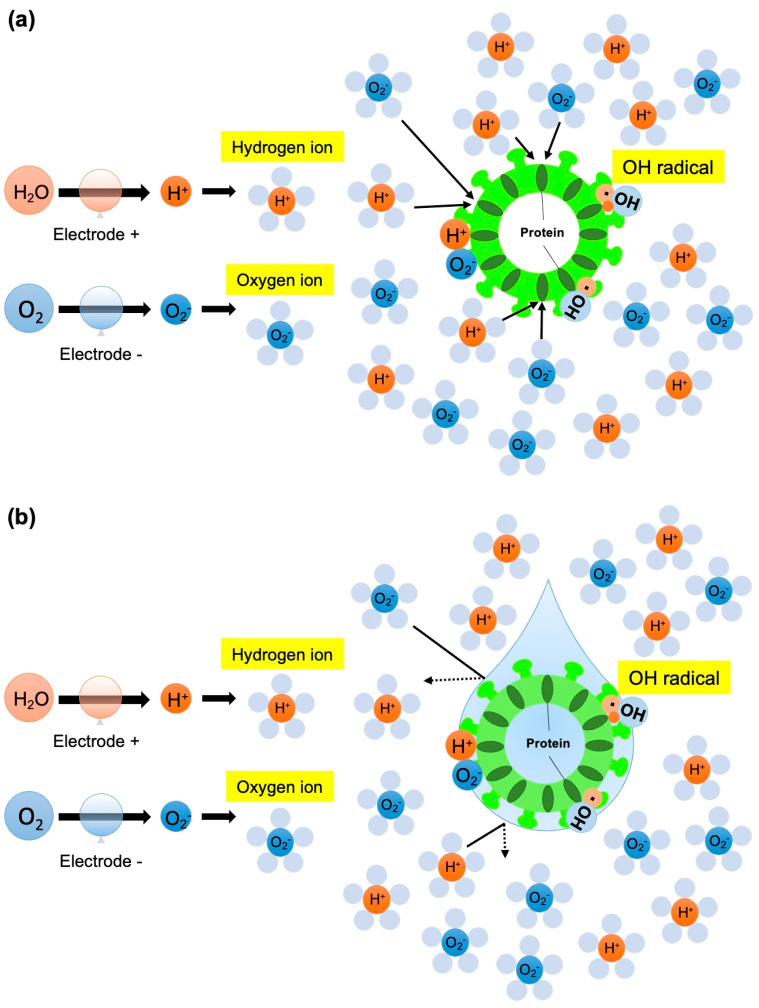
Schematic effect of ion irradiation on proteins on the surface of SARS-CoV-2 virion without water (**a**) or with water or saliva (**b**).

**Figure 3 viruses-16-00520-f003:**
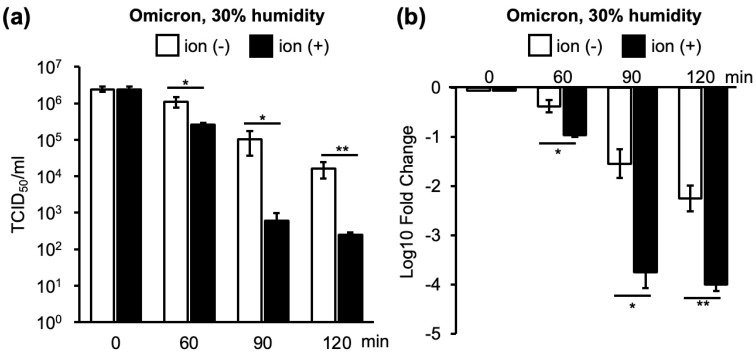
Illustration of standardization of results from inactivation tests using many viral strains with different infection titers. (**a**) TCID_50_ values of the Omicron strain after 60, 90, and 120 min of ion irradiation in a 30% humidity environment. (**b**) Log 10-fold changes in infectivity titers calculated using the TCID_50_ value just before ion irradiation as the standard. Both (**a**,**b**) graphs demonstrate a decrease in the infectivity of the Omicron strain over time, with further reduction observed upon ion irradiation. ‘ion (−)’ denotes samples without ion irradiation, while ‘ion (+)’ indicates samples subjected to ion irradiation. The reduction in infectivity titers is compared between samples with and without ion irradiation, and significant differences are assessed using Student’s *t*-test (* *p* < 0.05, ** *p* < 0.01).

**Figure 4 viruses-16-00520-f004:**
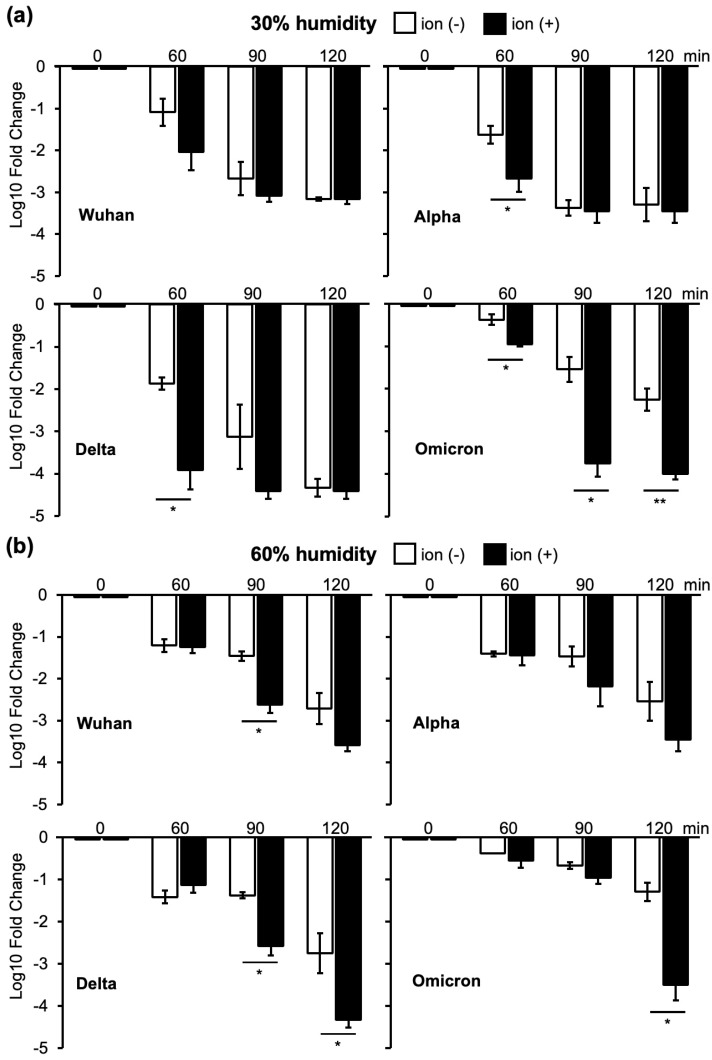
Log 10-fold changes in infectivity titers calculated using the TCID_50_ value just before ion irradiation as the standard. Four VOCs were subjected to 60, 90, and 120 min of ion irradiation. (**a**): Wuhan strain (left top), Alpha strain (right top), Delta strain (left second), and Omicron strain (right second) in a 30% humidity environment. (**b**): Wuhan strain (left top), Alpha strain (right top), Delta strain (left second), and Omicron strain (right second) in a 60% humidity environment. ‘Ion (−)’ denotes samples without ion irradiation, while ‘Ion (+)’ indicates samples subjected to ion irradiation. The reduction in infectivity titers is compared between samples with and without ion irradiation, and significant differences are assessed using Student’s *t*-test (* *p* < 0.05, ** *p* < 0.01).

**Figure 5 viruses-16-00520-f005:**
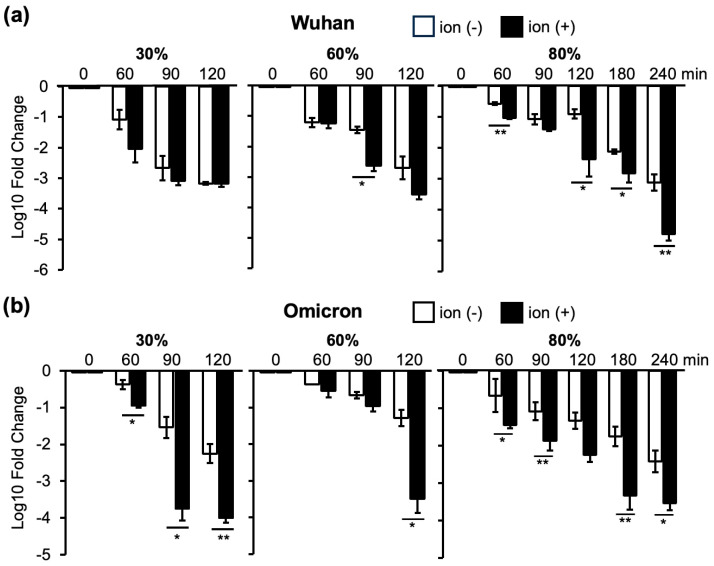
Log 10-fold changes in infectivity titers calculated using the TCID_50_ value just before ion irradiation as the standard. Changes in the Wuhan strain (**a**) and the Omicron strain (**b**) in 30%, 60%, and 80% humidity conditions. ‘Ion (−)’ denotes samples without ion irradiation, while ‘Ion (+)’ indicates samples subjected to ion irradiation. The reduction in infectivity titers is compared between samples with and without ion irradiation, and significant differences are assessed using Student’s *t*-test (* *p* < 0.05, ** *p* < 0.01).

**Figure 6 viruses-16-00520-f006:**
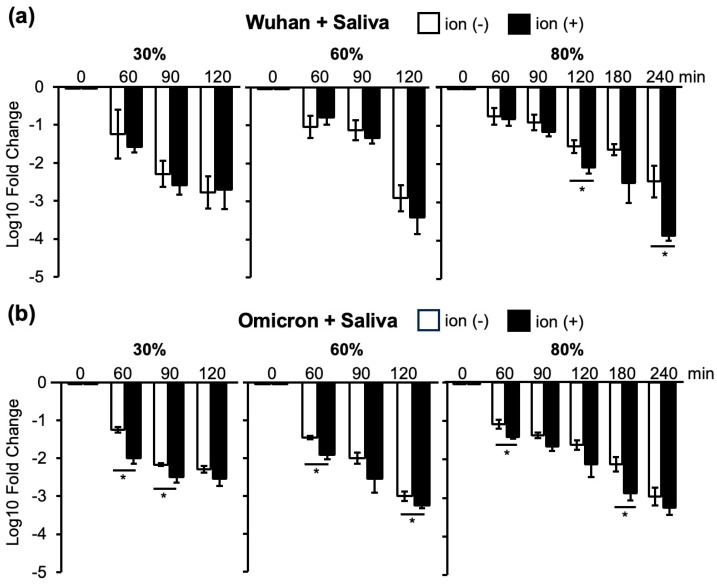
Log 10-fold changes in infectivity titers calculated using the TCID_50_ value just before ion irradiation as the standard. Changes in the Wuhan strain with saliva (**a**) and the Omicron strain with saliva (**b**) in 30%, 60%, and 80% humidity conditions. ‘Ion (−)’ denotes samples without ion irradiation, while ‘Ion (+)’ indicates samples subjected to ion irradiation. The reduction in infectivity titers is compared between samples with and without ion irradiation, and significant differences are assessed using Student’s *t*-test (* *p* < 0.05).

## Data Availability

The published article includes all datasets generated or analyzed during this study.

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
