# Peer review of "Effects of Strain Differences, Humidity Changes, and Saliva Contamination on the Inactivation of SARS-CoV-2 by Ion Irradiation"

_viruses, 2024, doi:10.3390/v16040520_

Round 1
Reviewer 1 Report
Comments and Suggestions for Authors
To the Authors,
The manuscript by Madi and colleagues discusses the effect of humidity and saliva in the inactivation of different SARS-CoV-2 strains through ion irradiation.
Despite the lack of any representative image of the reduced viral titers upon treatment, and the use of the incorrect statistical analysis tests, the manuscript is fluid and well written.
I notice a lack of future perspective in this manuscript, as the Authors did not propose neither a future application for the discoveries they made, nor a modification of the current approaches used for viral-positive samples handling.
Below are my points, listed as major and minor, which should be addressed by the Authors to make this manuscript suitable for publication.
Major
- In addition to bar plots shown throughout the text, the Authors must show representative images of the plates they used to observe CPE and from which they calculated TCID50 values.
- The statistical analysis and notation is completely incorrect. The Authors have used Student’s t-test to compare the difference between the means of more than two groups, which is wrong (even if they have compared them two by two). They must use ANOVA and appropriate post-hoc test as this is the proper way to analyze more than two groups. Furthermore, the Authors must report the notation “not significant – ns” when appropriate.
- What are the Authors showing in Figure 3, log fold-change of what? Are they showing TCID50 values as done in Figure 2? Please specify in the text and in the Figure legend.
- Again, what are the Authors showing in Figure 5 and 6?
- How the Authors explain the different behavior of Omicron strains compared to the other tested in this manuscript? As the currently circulating variants are all Omicron-related, it would be of interest to discuss how and why this strain is differently sensitive to humidity and saliva compared to the others.
- Which is the applicability of the discoveries the Authors report in this manuscript?
How the Authors recommend that the information about the different resistance to ion irradiation can be applied in a medical context? How can these informations have resonance in the medical and non-medical procedures yet in place?
Please discuss these points in the text.
Minor
- In the Abstract the Authors should specify that when referring to “virus” the Authors mean SARS-CoV-2; accordingly, in the first line of the Results section it should be specified that Omicron is a SARS-CoV-2 variant.
- Be consistent while describing figure legends and with Figure subpanels notation; for example, Figure 3 and 4 show the same type of experiment but the captions are way different; in the discussion, figure subpanels are reported in uppercase, while in the rest of the manuscript in lowercase; sometimes figures are indicated as “Fig.” and sometimes with “Figure”.
- The Authors state that: “By using the ozone catalyst, the ozone concentration around the virus-loaded membrane was kept to 0.05 ppm at all humidity conditions and never exceeded 0.1 ppm.”, did they measure the ozone levels? If yes, how?
- It is hard to compare the ozone effect on SARS-CoV-2 strains and bacteria (Staphylococcus); the Authors can mention the bactericidal activity of the ozone, but they have to discuss the use of ozone as antiviral.
- I recommend to show the Figure 7 early in the text, as this would help the reader to understand the treatment the Authors perform. The could consider to make Figure 7 as Figure 1b.
Typos
- Check typos in the sentence in page 2 from line 57 to 61.
- At page 7 line 231, page 8 line 244 and page 8 line 251what the Authors mean for “containment”? Do they mean “content” or “contamination”?
- In the caption of Figure 7 please correct “SARS-C0V-2” in “SARS-CoV-2”.
English is fine, minor check required.
Author Response
Major
- In addition to bar plots shown throughout the text, the Authors must show representative images of the plates they used to observe CPE and from which they calculated TCID50 values.
Response: We would like to thank for the valuable comments. A photo of the plate used to determine the TCID values shown in Figure 3a has been added to Figure S1.
- The statistical analysis and notation is completely incorrect. The Authors have used Student’s t-test to compare the difference between the means of more than two groups, which is wrong (even if they have compared them two by two). They must use ANOVA and appropriate post-hoc test as this is the proper way to analyze more than two groups. Furthermore, the Authors must report the notation “not significant – ns” when appropriate.
Response: When comparing between two groups with and without ions in Fig. 4, 5, and 6, we used Student's t-test. In this case, lines were drawn between the data being compared and the comparisons were clearly indicated. On the other hand, since comparisons involving four or three VOCs were also present in Fig. 4, 5, and 6, we conducted comparisons using One-way ANOVA followed with Tukey's multiple comparison test. The p-values obtained fron the Turkey’s multiple comparison test was shown in Fig. S2 and Fig. S3.
- What are the Authors showing in Figure 3, log fold-change of what? Are they showing TCID50 values as done in Figure 2? Please specify in the text and in the Figure legend.
Response: Thank you for your clarification. In each of Figures 4 (Fig. 3 in the former version) amd Figure 5 (Fig. 4 in the former version), we have specified in the text and figure legends what changes in the results are being represented.
- Again, what are the Authors showing in Figure 5 and 6?
Response: Thank you for your update. In Figures 6 and 7 (Fig. 5 and 6 in the former version) as well, we have specified in the text and figure legends what the graphs represent.
- How the Authors explain the different behavior of Omicron strains compared to the other tested in this manuscript? As the currently circulating variants are all Omicron-related, it would be of interest to discuss how and why this strain is differently sensitive to humidity and saliva compared to the others.
Response: Thank you for your valuable input. It has been reported that the Omicron variant contains numerous mutations in the viral structural proteins compared to the conventional strains. As mentioned in lines 350-362 of the Discussion, it has been reported that the shape of the spike protein of the Omicron variant is different from that of other VOCs. Furthermore, different groups reported that the Omicron variant can maintain infectivity for a longer period. We have mentioned this aspect and added references 29 and 30 to support our discussion.
- Which is the applicability of the discoveries the Authors report in this manuscript?
How the Authors recommend that the information about the different resistance to ion irradiation can be applied in a medical context? How can these informations have resonance in the medical and non-medical procedures yet in place? Please discuss these points in the text.
Response: Thank you very much for your valuable input. We believe that not only in healthcare settings but also in places where people gather such as department stores, movie theaters, performance venues, and indoor concert halls, it is important to consider measures to suppress airborne transmission. We have discussed this in the manuscript from line 308 to line 317.
Minor
- In the Abstract the Authors should specify that when referring to “virus” the Authors mean SARS-CoV-2; accordingly, in the first line of the results section it should be specified that Omicron is a SARS-CoV-2 variant.
Response: We thank the reviewer for the valuable comment. We have specified that Omicron is a SARS-CoV-2 variant in the first line of the Results section (3.1).
- Be consistent while describing figure legends and with Figure subpanels notation; for example, Figure 3 and 4 show the same type of experiment but the captions are way different; in the discussion, figure subpanels are reported in uppercase, while in the rest of the manuscript in lowercase; sometimes figures are indicated as “Fig.” and sometimes with “Figure”.
Response: We thank the reviewer for the valuable comment. As pointed out, the captions are very different between Figures, so we have fixed as shown in Figure 4, 5, and 6. We have unified to indicate “Figure” and stopped to use “Fig.”.
- The Authors state that: “By using the ozone catalyst, the ozone concentration around the virus-loaded membrane was kept to 0.05 ppm at all humidity conditions and never exceeded 0.1 ppm.”, did they measure the ozone levels? If yes, how?
Response: Thank you for your feedback. We have included the ozone measurement apparatus in the Materials and Methods section, and we have presented the measurement results in Supplementary Figure 5.

Reviewer 2 Report
Comments and Suggestions for Authors
In the article titled, “Effect of Strain Differences, Humidity Changes, and Saliva Contamination on Inactivation of SARS-CoV-2 by Ions” the authors show that SARS-CoV-2 VOCs have differential sensitivity to inactivation by humidity, saliva contamination, and ions.
Concerns:
1. Several panels in this paper are used multiple times throughout the manuscript.
a. Fig. 2 right panel is reproduced in Fig. 3a bottom right, Fig. 4b Left, Fig. 6a left
b. Fig 3a top left is reproduced in Fig. 4a left
c. Fig. 3b top left is reproduced in Fig. 4a middle
d. Fig 3b bottom right is reproduced in Fig 4b middle and 6a middle
e. Fig 5b is reproduced as Fig 6b (all 3 panels)
If you need to make that many comparisons to the same data, you should use independent controls for each experiment or find a way to summarize the data in such a way as it can be compared – a summary table might help with this.
2. Statistics – be clearer about what the comparisons were. I think it was ion vs no-ion. Figure legends should tell what statistics were used to compare what variable and what each asterisk p value cutoff is. Additionally, please indicate the number of replicates for each condition and whether they were technical or biological replicated.
3. Experimental details for ion treatment:
a. How much virus did you start with in TCID50? Was it consistent for each VOC?
b. How was the virus collected/separated from the membrane?
c. What fold serial-dilution was done to determine infectivity?
d. When adding saliva, did the total volume of virus on the membrane change? If you were using 50ul for no saliva, was the with saliva volume also 50ul? If not, drop volume may have impacted “drying out”.
4. Data visualization – it is very difficult to understand higher bars as being less virus. I highly recommend displaying the Log10 Fold change and start the y-axis at 0 with bars going down.
5. Additionally, comparing things in multiple panels is very difficult to assess so raw numbers should be put into the text as shown below in yellow. For instance (lines 184-185), inactivation of Omricon strain by ion irradiation was less efficient at 60% (5x10-3) humidity that at 30% humidity (1x10-4).
6. Comparing data sets. If you want to compare 30% humidity to 60% or 80%, do the statistical analysis to determine if they are actual different. Same with plus/minus saliva. Otherwise the conclusions you make about these comparison is much to strong.
7. There is a formatting issue with Figure 4 where the legend is not linked to the figure.
8. Line 311-312: what is the basis of the claim that Omicron is not sensitive to inactivation? Was this in reference to the study described in the previous lines? All data in this paper suggests that omicron is sensitive to inactivation. It may be less sensitive and need more time than other VOCs but eventually, there is inactivation.
9. Lines 318-319: This is not an accurate conclusion. There are many instances where there is no statistical difference between the ion and no-ion treatments (Figure 3).
Comments on the Quality of English LanguageThere was mostly formatting issues with the English language. For instance, superscripts and subscripts were not used in the test. Certain words need to be italicized. The issues would be easily caught by a the copy editor.
Author Response
Concerns
- Several panels in this paper are used multiple times throughout the manuscript.
- Fig. 2 right panel is reproduced in Fig. 3a bottom right, Fig. 4b Left, Fig. 6a left
- Fig 3a top left is reproduced in Fig. 4a left
- Fig. 3b top left is reproduced in Fig. 4a middle
- Fig 3b bottom right is reproduced in Fig 4b middle and 6a middle
- Fig 5b is reproduced as Fig 6b (all 3 panels)
If you need to make that many comparisons to the same data, you should use independent controls for each experiment or find a way to summarize the data in such a way as it can be compared – a summary table might help with this.
Response: We thank the reviewer for the valuable comment. As noted, a significant portion of the data has been reproduced, stemming from the approach of constructing the paper by extracting and comparing individual data points. To address the limitations of this approach and offer a comprehensive overview of the dataset, we have introduced a heat map for comparing fold changes in infectivity instead of a summary table, presented as Supplemental Figure 4.
- Statistics – be clearer about what the comparisons were. I think it was ion vs no-ion. Figure legends should tell what statistics were used to compare what variable and what each asterisk p value cutoff is. Additionally, please indicate the number of replicates for each condition and whether they were technical or biological replicated.
Response: Thank you very much. As suggested, we have added asterisks to indicate statistical significance when comparing ions versus non-ions and observing statistical differences. Also, we described in the legends of Figure 3, 4, 5, and 6 what statistics were used to compare what variable and what each asterisk p value cutoff is. We also indicated the number of replicates for each condition and whether they were technical or biological replicated.
- Experimental details for ion treatment:
- How much virus did you start with in TCID50? Was it consistent for each VOC?
Response: Since the TCID50 values vary for each strain, we have included the TCID50 values for each strain in the Materials and Methods section (from line 106 to line 113).
- How was the virus collected/separated from the membrane?
Response: We have immeresed the membrane in the culture medium to collect samples.
- What fold serial-dilution was done to determine infectivity?
Response: We have described in the Materials and Methods, section 2.3 Virus titration (from line 117 to line 119). Viruses are subjected 3.16-fold serial dilution before inoculation to cells.
- When adding saliva, did the total volume of virus on the membrane change? If you were using 50 μL for no saliva, was the with saliva volume also 50 μL? If not, drop volume may have impacted “drying out”. 
Response: We have carefully adjusted the volume of virus fluid to 50 μL in both cases, with and without saliva. Additionally, we have ensured that both preparations contain the same TCID50.
- Data visualization – it is very difficult to understand higher bars as being less virus. I highly recommend displaying the Log10 Fold change and start the y-axis at 0 with bars going down.
Response: Thank you very much. As suggested, we have changed the display of the Log10 Fold change to start he y-axis at 0 with bars going down in Figures 3, 4, 5, and 6.
- Additionally, comparing things in multiple panels is very difficult to assess so raw numbers should be put into the text as shown below in yellow. For instance (lines 184-185), inactivation of Omricon strain by ion irradiation was less efficient at 60% (5x10-3) humidity that at 30% humidity (1x10-4).
Response: Thank you very much. In the revised manuscriot, we have put the fold change of TCID 50 in the parenthesis in the text from line 217 to line 220.
- Comparing data sets. If you want to compare 30% humidity to 60% or 80%, do the statistical analysis to determine if they are actual different. Same with plus/minus saliva. Otherwise the conclusions you make about these comparison is much to strong.
Response: Thank you for your valuable input. Because comparisons involving four or three VOCs were also present in Fig. 4, 5, and 6, we conducted comparisons using One-way ANOVA followed with Tukey's multiple comparison test. The p-values obtained fron the Turkey’s multiple comparison test was shown in Fig. S2 and Fig. S3.
- There is a formatting issue with Figure 4 where the legend is not linked to the figure.
Response: Thank you for pointing out the formatting error. We have corrected the error.
- Line 311-312: what is the basis of the claim that Omicron is not sensitive to inactivation? Was this in reference to the study described in the previous lines? All data in this paper suggests that omicron is sensitive to inactivation. It may be less sensitive and need more time than other VOCs but eventually, there is inactivation.
Response: We would like to thank for the valuable comments. While the Omicron variant virus undergoes natural decay and further inactivation due to ion irradiation, we believe that the results from Figures 4 and 5 indicate that Omicron is not particularly sensitive to inactivation compared to other strains. Furthermore, as stated in the Discussion (from line 350 - 354), two different research groups have shown research findings suggesting that Omicron is not particularly sensitive to inactivation compared to other strains.
- Lines 318-319: This is not an accurate conclusion. There are many instances where there is no statistical difference between the ion and no-ion treatments (Figure 3).
Response: Upon reviewing the results presented in the new Figure 4, we have noticed that in some cases, there are no statistically significant differences observed between the ion-treated and non-ion-treated samples. We have revised the statement to indicate that the description now aligns with the results obtained from t-tests and multiple comparisons, focusing on instances where significant differences were indeed observed (from line 199 – 201, and from line 211 - 222). We have accordingly modified the discussion (from line 368 – 370) according to the revised results.
We appreciate your attention to detail and value your feedback in ensuring the clarity and integrity of our research findings.
Comments on the Quality of English Language:
There was mostly formatting issues with the English language. For instance, superscripts and subscripts were not used in the test. Certain words need to be italicized. The issues would be easily caught by a the copy editor.
Response: Thank you very much for valuable notification. We have carefully examined formatting issue and corrected superscripts and subscripts. We have also checked italicized description.

Reviewer 3 Report
Comments and Suggestions for Authors
Afifah Fatimah Azzahra Ahmad Wadi, Hironori Yoshiyama and colleagues present an interesting study on the effects on inactivation of SARS-CoV-2 by ions. The study requires several clarifications and additional quantification. In addition, the manuscript should be enriched with the topic of the present study. Several problems are summarized in the following points; they greatly weaken the study:
1 In the abstract, the author draws out the significance of the study of this manuscript in only one sentence, which is suggested to be complemented. And the abstract should end with a conclusion to the research.
2 Line 57. The logical relationship in this paragraph is not clear enough. Since indoor humidity promotes the outbreak of SARS-CoV-2, it is too far-fetched to study the causal relationship of humidity affecting ionizing radiation against the virus.
3 There are too many dotted lines in Figure 1, which makes the picture confusing and difficult to understand, so it is recommended that the dotted lines be removed and a graphical note be added to show that Pump and Hygrometer are Humidity controllers.
4 I noticed that in the culture of Vero cells, the authors added 1 mg/mL G418, which should only be required for antibiotic screening and not for normal culture of Vero cells, whereas the description in the manuscript would lead one to believe that the addition of G418 is required for the culture of Vero cells in any case.
5 Figure 5b and Figure 6b look the same both in terms of the method and the results, what is the difference between them, and if the results are the same, please don't repeat them.
Comments on the Quality of English LanguageSome of the following minor mistakes also appear several times in the manuscript, so please check twice and revise them.
Line 80 CO2 should be CO2. please revise the similar problems (such as Line 84 104 should be 104. Line 107 TCID50 should be TCID50) together.
Line 100 3.14 -fold series dilution of virus should be converted to detailed viral titers.
Line 128 Extra spaces before % should be removed.
Line 155 p < 0.05 should be italicized. And the differences of * and ** should be depicted clearly.
Line 157 I suggest that the left and right in Figure 2 should be changed to a and b.
Line 219 indicate should be indicated.
Line 270 SARS-COV-2 should be SARS-CoV-2.
Author Response
- In the abstract, the author draws out the significance of the study of this manuscript in only one sentence, which is suggested to be complemented. And the abstract should end with a conclusion to the research.
Response: Thank you for pointing out. I have added a conclusion at the end of the summary.
- Line 57. The logical relationship in this paragraph is not clear enough. Since indoor humidity promotes the outbreak of SARS-CoV-2, it is too far-fetched to study the causal relationship of humidity affecting ionizing radiation against the virus.
Response: Thank you very much. We modified the statement as follows; The efficiency of virus inactivation by ions is influenced by indoor environment. Indoor relative humidity is expected to modulate the occurrence of COVID-19 [16]. Moderate humidity has been reported to have a better association with the COVID-19 outbrakes [17]. We considered ionization as a means to disinfect viruses in the environment. However, ions undergo chemical reactions upon contact with water, hence, it is necessary to investigate how humidity affects the effectiveness of ion irradiation against SARS-CoV-2 (lines 62-67).
- There are too many dotted lines in Figure 1, which makes the picture confusing and difficult to understand, so it is recommended that the dotted lines be removed and a graphical note be added to show that Pump and Hygrometer are Humidity controllers.
Response: Thank you for your advice. We have corrected as pointed out.
- I noticed that in the culture of Vero cells, the authors added 1 mg/mL G418, which should only be required for antibiotic screening and not for normal culture of Vero cells, whereas the description in the manuscript would lead one to believe that the addition of G418 is required for the culture of Vero cells in any case.
Response: I apologize. We are using G418 to maintain the expression of TMPRSS2 in VeroE6 cells. However, G418 affects cell proliferation. Therefore, cells infected with SARS-CoV-2 are first cultured in G418-free media to allow proliferation, followed by virus infection assays using G418-free media. This has been described and added to the Materials and Methods section (from line 91 to line 94).
- Figure 5b and Figure 6b look the same both in terms of the method and the results, what is the difference between them, and if the results are the same, please don't repeat them.
Response: Thank you very much for your valuable input. In the initial manuscript, Fig. 5b and Fig. 6b were the same figure. However, in the revised manuscript, we have made changes. Figure 6(a) now represents the Wuhan strain suspended in saliva, while Figure 6(b) represents the Omicron strain suspended in saliva. Additionally, the Omicron strain suspended in media is now depicted only in Figure 5(b).
Comments on the Quality of English Language: Some of the following minor mistakes also appear several times in the manuscript, so please check twice and revise them.
Line 80 CO2 should be CO2. please revise the similar problems (such as Line 84 104 should be 104. Line 107 TCID50 should be TCID50) together.
Response: The “CO2” has been corrected to “CO2” on line 88. Similarly, 104 was corrected to 104 on line 95, and TCID50 on line 107 was corrected to TCID50 on line 124. We have carefully examined and corrected superscripts and subscripts throughout the manuscript.
Line 100 3.14 -fold series dilution of virus should be converted to detailed viral titers.
Response: I apologize for the mistake. It should have been 3.16 instead of 3.14. The square of 3.16 is approximately 10. When determining the infectivity titer, we perform 12 repetitions of 3.16-fold dilutions, allowing us to observe a difference of 10^6-fold. This explanation has been added to line 118.
Line 128 Extra spaces before % should be removed.
Response: We have corected as seen on line 145 in the revised manuscript.
Line 155 p < 0.05 should be italicized. And the differences of * and ** should be depicted clearly.
Response: Line 184 p < 0.05 has been changed to italicize. Moreover, * and ** indicate p < 0.05 and p < 0.01, respectively.
Line 157 I suggest that the left and right in Figure 2 should be changed to a and b.
Response: Thank you for your feedback. The footnote for Figure 2 has been corrected to indicate Figure 2a and Figure 2b.
Line 219 indicate should be indicated.
Response: We have corected as seen on line 247 in the revised manuscript.
Line 270 SARS-COV-2 should be SARS-CoV-2.
Response: We have corected as seen on line 302 in the revised manuscript.

Round 2
Reviewer 1 Report
Comments and Suggestions for Authors
I would like to thank the Authors for addressing my concerns properly.
I do appreciate the tone and attitude of their comments,
I strongly recommend this manuscript for publication in Viruses.
Reviewer 2 Report
Comments and Suggestions for Authors
I appreciate the author's willingness to perform to all of the request in my original review. I have no further suggestions!